# Death-Leading Envenomization of Rabbits with Snake Versus Scorpion Venoms: A Comparative Forensic Investigation of Postmortem Decomposition and Beetle Succession

**DOI:** 10.3390/insects16060625

**Published:** 2025-06-13

**Authors:** Afnan Saleh Al-Qurashi, Mohammed Saleh Al-Khalifa, Hathal Mohammed Al Dhafer, Mahmoud Saleh Abdel-Dayem, Hossam Ebaid, Ashraf Mohamed Ahmed

**Affiliations:** 1Zoology Department, College of Science, King Saud University, Riyadh 11451, Saudi Arabia; 443203299@student.ksu.edu.sa (A.S.A.-Q.); mkhalifa@ksu.edu.sa (M.S.A.-K.); habdrabou@ksu.edu.sa (H.E.); 2Plant Protection Department, College of Food and Agriculture Sciences, King Saud University, Riyadh 11451, Saudi Arabia; hdhafer@ksu.edu.sa (H.M.A.D.); mseleem@ksu.edu.sa (M.S.A.-D.)

**Keywords:** beetles, corpse, decomposition, envenomization, forensic, snake, scorpion, venoms

## Abstract

This study investigates the impact of antemortem envenomization of rabbits by snake versus scorpion venom on the postmortem decomposition process and the succession pattern of the associated beetles. The results revealed a venom-dependent impact on the decomposition process and beetle succession. The decomposition duration was prolonged for up to one day. The succession pattern and quantitative and qualitative analysis of the beetle community varied significantly between the treated corpses and the decomposition stages. This could be of forensic importance since envenomation with venomous animals is a considerable death-leading cause worldwide.

## 1. Introduction

Forensic entomology is a rapidly growing field of study that involves the use of insects to aid in criminal investigations. In this field, necrophagous insects and other arthropods are used as tools of forensic investigation to detect, elucidate, and establish evidence during forensic investigation [1]. This is due to the capability of these insects to reach a corpse within minutes post-death [2]. The diversity of these corpse-seeking insects, their successional behavior towards the corpse, the relationship between their arrival time and the developmental process of their immature stages, and their feeding manner on the corpse all provide valuable data to forensic entomologists that help them solve crime aspects [3,4,5] and estimate the post-mortem interval (PMI) [6].

There are four groups of corpse-seeking insects, which are categorized based on their feeding manner [2]. The first group is sarcosaprophages, which include dipteran flies from Calliphoridae, Muscidae, and Sarcophagidae, as well as coleopteran beetles from the Dermestidae Histeridae, Silphidae, and Staphylinidae families, which feed on decomposing corpses. The second group is coprophages, which include beetles from Scarabaeidae and flies from Calliphoridae, Sarcophagidae, and Muscidae, which are attracted to the herbivores’ rumen contents. The third group is dermatophages, which include beetles from Dermestidae, which feed on cadaver remnants, such as bones, hair, and dried skin. The fourth group is predaceous insects, such as Staphylinid and Histerid beetles and Formicid ants, which feed mainly on corpse-colonizers like dipteran larvae. These insects colonize the corpse along with microorganisms, where they work to decompose the corpse [7]. During decomposition, the corpse goes through five stages (fresh, bloating, active decay, advanced decay, and remain), where each stage is characterized by a specific attraction of insects [8].

Necrophagous beetles constitute a forensically determinant factor due to their ability to inhabit the majority of the corpse environment, offering valuable data for forensic investigation [9]. They belong to Coleoptera, the largest Order comprising over one-third of all known insect species [10]. These corpse-associated insects provide valuable data that help estimate the PMI of dried corpses as well as evaluate the damage and variations that may have occurred to the corpse status [8]. The most forensically important beetle families include Cleridae, Dermestidae, Histeridae, Silphidae, Staphylinidae, and Scarabaeidae [8]. Nevertheless, much less research on corpse-seeking insects has given attention to these beetles compared to flies [11,12,13,14]. Therefore, the present study was implemented to participate in compensating for this deficiency.

Envenomation-related death cases constitute a vital and considerable aspect from the forensic point of view. This is based on the fact that venomous snakes and scorpions are distributed all over the world [15,16,17,18,19,20,21,22,23,24] and affect millions of people [24,25,26,27,28,29,30,31,32,33]. Every year, snakebites cause a staggering number of deaths around the world—somewhere between 81,000 and 138,000 people lose their lives, according to the World Health Organization. On top of that, as many as 400,000 people who survive these bites are left with lasting disabilities [34]. Scorpion stings are a major health issue worldwide, causing around 1.2 million stings and more than 3000 deaths each year [30]. Consequently, envenomation is considered a significant cause of death worldwide [16,24,35,36,37,38]. In this regard, the majority of envenomation-related deaths are caused by arthropods [36,37,39], followed by snakes [24,26]. In regions like Saudi Arabia, snakebites [33,40,41] and scorpion stings [40,42,43,44] are prevalent, with incidence rates of 10 to 60 cases per 100,000 population annually for snakebites [33] and up to 5000 scorpion stings reported per year [41,42]. These statistics show just how important it is to understand what happens to the body after a venomous bite or sting—especially when it comes to forensic investigations. The desert black snake, *Walterinnesia aegyptia* L., and the fat-tail scorpion, *Androctonus crassicauda* L., are significant contributors to envenomation-related deaths, yet there is a lack of research on how their venoms affect the decomposition process and beetle succession. Thus, this study aims to bridge this gap by comparing the effects of *W. aegyptia* and *A. crassicauda* venoms on rabbit corpses, focusing on the decomposition stages and beetle succession patterns. By addressing this knowledge gap, this research seeks to enhance PMI estimation and provide valuable insights into differentiating between causes of death involving venomous animals, ultimately contributing to the advancement of forensic science in regions with high envenomation rates.

## 2. Materials and Methods

### 2.1. Meteorological Parameters

The experiments of this study ran for 11 days in the summer (from 6–17 June 2023). The atmospheric parameters were determined over the study period via the Saudi National Center for Meteorology [45]. In addition, the on-site atmospheric parameters were also reported manually at the experimental site in a daily manner (at midday) during the study period following [46,47]. Relative humidity (RH) and ambient temperature (°C) were measured using a hygrometer and digital thermometer devices (Elitch Technology, Nanjing, Jiangsu, China) following their instruction manuals. The Skywatch Wind Meter device (Skywatch^®^, Yverdon-les-Bains, Switzerland) was used for monitoring wind speed following its instruction manual.

### 2.2. Experimental Site

The experiments were carried out in the botanical garden of the Botany and Microbiology Department, College of Science, King Saud University, Riyadh, Saudi Arabia (24°43.174′ N, 46°36.954′ E) (Figure 1A). This garden covers an area of about 10,000 m^2^ with a clayey sandy soil with abundant grasses and herbs. It houses a significant number of plant species, ranging from wild, medicinal, and endangered plants to economically important plants [48]. Although this site does not fully represent typical forensic crime scenes such as urban or peri-urban settings, it has been targeted in this study as it offers a standardized and semi-natural habitat with a plant-rich environment that reflects ecological realities encountered in field-based forensic scenarios that we assume forensic investigations are likely to take place in.

### 2.3. Experimental Animals

#### 2.3.1. Rabbits and Mice

Swiss albino mice (≈18–20 g each) were obtained from the Animal House, Zoology Department, College of Science, King Saud University, for carrying out the lethality of snake and scorpion venoms following [49]. Male domestic rabbits, *Oryctolagus cuniculus domesticus* (Linnaeus, 1758) (≈2.8–3.0 kg each), were purchased at a specialized local farmer’s market in Riyadh City, Saudi Arabia. They were used as experimental models for conducting the forensic experiments in this study according to [50] and following other studies [47,51,52,53,54]. Prior to use for experiments, the rabbits were first kept for acclimatization for 2 days in the animal house. They were housed in standard rearing conditions with suitable feed and aeration by skillful specialists in accordance with the Research Ethical Committee at King Saud University (approval code: KSU-SE-23-83, approval date: 14 September 2023).

#### 2.3.2. Snakes and Scorpions

The Saudi desert black cobra snake, *W. aegyptia* Lataste, 1887 (Squamata, Elapidae) [18,55,56,57] (Figure 1B; from [58]), and the fat-tail scorpion, *A. crassicauda* (Olivier, 1807) (Scorpiones, Buthidae) [21,59,60,61] (Figure 1C; from [62]) were from the Animal House, Zoology Department, College of Science, King Saud University and used for carrying out this study. The snakes and scorpions were housed in standard laboratory conditions in the animal house with feed and aeration, according to [63,64], respectively. The snakes and scorpions were handled by a qualified specialist from the Herpetology laboratory in the Zoology Department and in accordance with the Research Ethical Committee at King Saud University (approval code: KSU-SE-23-83).

### 2.4. Collection of Venoms

The snake crude venom was milked from five adult snakes, according to [63,65]. Scorpion crude venom was collected from 100 scorpions by inducing electric shock according to [66] and as detailed in [64]. The collected samples of each venom were pooled, lyophilized, and stored at −20 °C until used. Prior to experimental use, an aliquot from each venom was freshly prepared in 0.9% saline (pH 7.2), according to [67], in a final concentration of 1.0 mg/mL (*w*/*v*). Carrying out venom collection took place by a qualified specialist from the Herpetology Laboratory in Zoology Department and in accordance with the Research Ethical Committee at King Saud University (approval code: KSU-SE-23-83).

### 2.5. Lethality Assay

The lethality, in terms of both LD_50_ and LD_95_, for each venom was determined following the World Health Organization’s guidelines [68] and as per detailed in our previously published study [49]. Briefly, a preliminary dose-finding experiment was first carried out using Swiss albino mice. The preliminary screening was carried out using eight descending dilutions from each venom aliquot (1000, 500, 250, 125, 62, 31, 15, and 7 µg/mL). Then, 0.2 mL from each concentration was subcutaneously injected into each of the eight mice (one mouse for each dilution). The mortalities were then recorded at 24 h post-envenomization. The range of each venom concentration was then narrowed to the required concentration for subsequently conducting the main full lethality assays.

Based on the preliminary screening of each venom lethality, the 4 actively effective concentrations of each venom that caused mice mortality ranging from 0.0 to 100% were used to carry out a full lethality test to determine the LD_50_ and LD_95_. In this experiment, 5 groups (one group for each concentration) of mice (5 mice/group; *n* = 5) were subcutaneously injected with the venom. The mortalities were recorded at 24 h post-envenomization, and both LD_50_ and LD_95_ were calculated by the Probit analysis according to [69] and following [70].

### 2.6. Envenomization of Rabbits

Upon determining the LD_95_ of each venom in mice, it was converted into the equivalent doses for rabbits according to [71] and following [49]. Fifteen rabbits were divided into three groups: A, B, and C (5 rabbits each; *n* = 5). Each animal in groups A and B was injected with 0.5 mL venom aliquot that contained the equivalent LD_95_ of the *W. aegyptia* snake (0.264 mg/rabbit) and *A. crassicauda* scorpion (10.064 mg/rabbit), respectively. The venoms were intravenously injected into the rabbits via the ear vein (Figure 1D) following [72]. The use of intravenous injection was chosen for its precision in delivering a controlled dose, although it does not perfectly mimic natural bites or stings. The envenomized rabbits were deceased within 10 to 20 min post-envenomization. In parallel, the control group (group C) was injected with 0.9% saline prior to euthanasia with CO_2_ according to [73] and following [47]. The handling and killing of the animals were carried out in accordance with the Research Ethical Committee at King Saud University (approval code: KSU-SE-23-83).

### 2.7. Experimental Design

Within a maximum of 1 h from their confirmed death, the rabbit corpses were immediately translocated to the experimental site. The corpses were individually placed in a metal cage (50 × 40 × 25 cm each) (Figure 1E) to protect them against predation, as detailed in [46,47]. The cages were placed a minimum of 10 m away from one another to provide isolated resources for corpse-seeking insects, as recommended by [74] and to reduce the effect of odor interference, which could affect insect attraction [49,75]. Three pitfall traps (10 cm in diameter, 5 cm in depth each) were placed adjacent to each corpse (Figure 1E) to collect the attracted insects beyond the collection times [76]. Each trap contained a solution composed of 250 mL of water, 5% soap powder, and 5% NaCl according to [46]. Since direct sunlight impacts insect succession compared to shadow [77], all experimental corpses were standardized by placing them in shady places underneath trees.

**Figure 1 insects-16-00625-f001:**
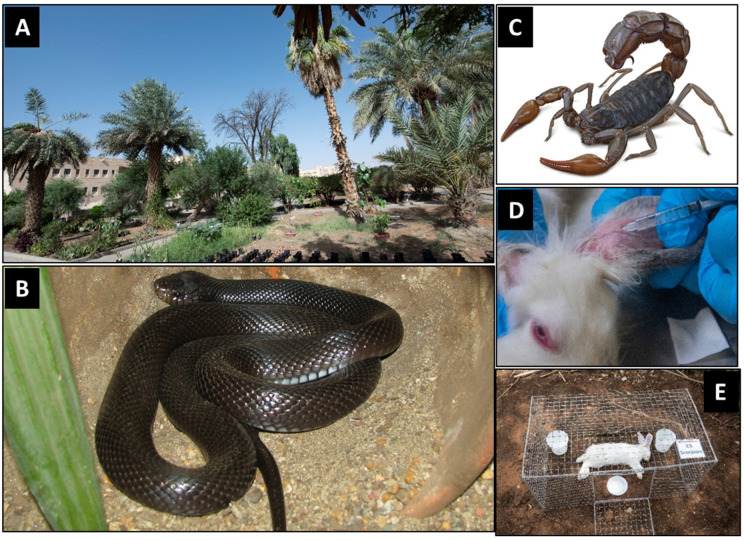
(**A**) Part of the botanical garden targeted by the study, (**B**) the desert black snake, *W. aegyptia* (from [58]), (**C**) the fat-tail scorpion, *A. crassicauda* (from [62]), (**D**) the ear intravenous injection of rabbits, and (**E**) an example of rabbit corpses in its protective metal cage along with the pitfall traps.

### 2.8. Decomposition Process

Each corpse was examined and investigated for 5 min in a daily manner, from the first day of the exposure until complete skeletonization, according to [49,78]. There are four to five stages that are commonly reported in the decomposition process [8]. Only four stages were distinctly observed in this study (fresh, bloated, decay, and dry stages) which reflect the actual decomposition process in our experimental conditions following [46,47,49]. The duration of each stage (in days) was reported until complete skeletonization (dryness).

### 2.9. Beetles Collection and Identification

Corpse-attracted beetles were collected from corpses on an hourly basis (2 min for each corpse) from 6 a.m. to 4 p.m. over the first three days of the experiment. Then, the insects were collected once a day at 6 a.m. until the end of the experiment according to [79] and following [46,49]. Only adults were included in the counting of the collected beetles during this study. The beetles were collected from and underneath the corpses using soft forceps and a spatula (3 cm in width and 10 cm in length) following [80]. Pitfall traps were also used not only to maintain monitoring beyond the time of collection but also to reduce the disturbance of carcasses-inhabitant insects for later sampling. The collected beetles were preserved in 70% ethanol and stored at 4 °C until used for counting and identification as detailed in [80]. The beetles were morphologically identified to the species level at the King Saud University Museum of Arthropods (KSMA) (https://cfas.ksu.edu.sa/en/node/3075, accessed on 2 May 2025) by the 3rd and 4th authors, H. Al-Dhafer and M. Abdel-Dayem, respectively. A small number of voucher specimens collected during this study have been preserved and deposited at KSMA.

### 2.10. Statistical Analysis

Mice-corrected mean lethality values were used to calculate the LD_50_ and LD_95_ of each venom using Probit Analysis following [69]. The resulting mean values from Probit analysis were considered significant (*p* < 0.5) if their 95% confidence limits (Upper to Lower) were not overlapped, according to [81,82]. The Minitab software (MINITAB, State College, PA, version 18.1, 2018, UK) was used to statistically analyze the results of the total counts of beetles. Prior to any further analysis, the normality of insects’ counts was tested using the Anderson–Darling Normality Test, according to [83]. Because the overall data of beetle counts were normally distributed, one-way ANOVA was used for the comparisons between the treated groups, and the Multiple Tukey’s Pairwise Comparison test was used to analyze the differences between means, according to [83]. However, data sets pertaining to the coleopteran families and species counts were not normally distributed and, thus, were analyzed using the non-parametric Mann–Whitney *U*-test. The data sets pertaining to the decomposition process showed timing identity of the reported durations of each stage for the 5 corpses (5 replicates) within each group. Thus, statistical analysis was not possible, and comparison between groups was considered based on the reported time period of each stage following [46,47,49]. Finally, singleton or doubleton was considered when only one or two individuals of a particular beetle species are reported, respectively, according to [84] and following [46,47,49,77]. All results are presented as the means of 5 replicates using five different individual rabbit corpses (*n* = 5) ± standard errors (SE), as determined by basic statistical analyses.

## 3. Results

### 3.1. Venoms Lethality

As shown in Table 1, Probit analysis revealed that LD_50_ of the *W. aegyptia* snake venom was 26.7 times higher than that of the *A. crassicauda* scorpion (*p* < 0.05), with non-overlapping confidence limits. In addition, the LD_95_ of *W. aegyptia* venom was 38.12 times higher than that of *A. crassicauda* (*p* < 0.05), with non-overlapping confidence limits. These LD_95_ values were converted into the equivalent (4×) for rabbits (0.264 and 10.064 mg/rabbit, respectively) and were considered the lethal doses used for rabbit envenomization.

### 3.2. Meteorological Measurements

#### 3.2.1. Atmospheric Parameters

The atmospheric parameters over the course of the 11-day experiment in Riyadh City are shown graphically in Figure 2. The maximum, minimum, and mean temperatures were constant (Figure 2A), with averages of 41.0, 29.27, and 35.14 °C, respectively (Figure 2B). The relative humidity (RH) was fluctuating from 5% during the first day to 13% during the last day of the experiment (Figure 2A), with an average of 7.8% (Figure 2B). The wind temperature was constant with an average of 32.03 °C, while the wind speed was fluctuating from 5 to 31 km/h during days 7 and 9, respectively (Figure 2A), with an average of 13.15 km/h (Figure 2B). These atmospheric parameters are within the natural range during this time of the year in Riyadh city [45].

#### 3.2.2. On-Site Recorded Weather Parameters

The on-site manually recorded weather parameters (Figure 2C,D) were fluctuating over the experimental period compared to the officially recorded ones by the Saudi National Center for Meteorology for Riyadh City (Figure 2A,B). The temperature recorded a minimum of 29 °C on day 6 and a maximum of 38 °C on day 4 (Figure 2C), with an average of 34 °C (Figure 2D). The relative humidity recorded a minimum of 30% and a maximum of 36% (Figure 2C), with an average of 32% (Figure 2D). Wind temperature recorded a minimum of 29 °C and a maximum of 34 °C (Figure 2C), with an average of 31.37 °C (Figure 2D). Fluctuations were noticeable in the on-site recorded temperature, RH, and wind temperature. The maximum fluctuation was recorded in wind speed, as most days recorded 0.0 km/h, while the maximum speed was 18 km/h on day 10 (Figure 2C), with an average of 3.25 km/h (Figure 2D).

### 3.3. Decomposition Stages

The four reported decomposition stages over the study period were fresh, bloating, decay, and dry (Figure 3). In all three treatments, rabbit corpses appeared in the fresh stage as if they were alive in terms of softness and flexibility at the beginning of death (Figure 3A). This stage lasted for 31 h in the control group, whereas it lasted for 21 h in both envenomized corpses (Table 2). It was noticeable that the snake- and scorpion-envenomized corpses became stiffened and somewhat rigid within 30 min and 6 h postmortem, respectively, and both began to emit slight unpleasant odors.

In the bloating stage (Figure 3B), the corpses started to swell from the abdominal area, progressing to the chest and neck, releasing offensive odor, but with different durations between treatments. This stage lasted for 10 and 24 h in snake- and scorpion-envenomized corpses, respectively, compared to 14 h in the control ones (Table 2). It was noticeable that the scorpion-envenomized corpses were releasing stronger offensive odor in this stage compared to the snake-envenomized and control ones.

The decay stage also varied among the three treatments (Figure 3(C1)). The control corpses showed distributed decomposition across body areas, partial appearance of dipteran larvae from body openings, strong offensive odors, fluid exudation, and lasted for about 48 h (Table 2). The snake-envenomized corpses started decomposition earlier, which was observed in the chest and neck areas. The abdominal contents were noticeably spreading around the corpse. There was significant fluid exudation, which lasted for approximately 62 h (Table 2). The scorpion-envenomized corpses started the decomposition at the same time as the control ones, and it was evident in the neck, chest, and forelimbs, with a stronger offensive odor. There were not as many corpse-colonizing larvae as in the control and snake-envenomized ones (Figure 3(C2)). The smell was also worse and stronger than in the other groups and lasted for about 72 h (Table 2).

In the dry stage, all corpse groups were characterized by complete dryness and rigidity (Figure 3D). This stage started at approximately 117 h (4.9 days) postmortem in the scorpion-envenomized corpses compared to 93 h (3.9 days) in both the control and snake-envenomized ones (Table 2).

### 3.4. Abundance of Corpse Associated Beetles

A total of 1094 corpse-attracted beetles were collected from all the experimental corpses during the entire experimental period of this study. Out of them, 36.2% (396) and 29.1% (319) of the beetles were collected from the snake- and scorpion-envenomized corpses, respectively, compared to 34.7% (379) collected from the control ones. As shown in Figure 4A, the number of beetles attracted to all corpses from the first day of exposure increased to the maximum during days 4 and 5, after which it reduced again to the minimum at the end of the experiment. The Interval Plot of beetles’ succession in each treatment versus the days postmortem over the experimental period (from day 1 to day 11) was calculated and created by the Pooled Standard Deviation, which revealed four major succession peaks (waves) of the attracted beetles to the corpses during the period from day 2 to day 7, with the highest peak on day 4 (Figure 4B).

When comparing the envenomized corpses over the 11-day experimental period, one-way ANOVA revealed significantly more beetles attracted to the snake-envenomized corpses on days 2 and 3, about the same number on day 4, and significantly fewer on day 5 compared to the scorpion-envenomized ones (*F*_32,132_ = 12.36, *p* < 0.05, *n* = 5) (Figure 4A). However, the scorpion-envenomized corpses attracted significantly more beetles later on day 5 (Figure 4A). Significant distinctive differences in beetle succession were observed between the snake- and scorpion-envenmized corpses in a venom-dependent manner.

### 3.5. Differential Abundance of Beetles

The impact of venom type on the differential abundance of attracted coleopteran families is represented in Figure 5A. Overall, the most abundant families were Histeridae, followed by Dermestidae, Scarabaeidae, and Tenebrionidae, while the least abundant ones were Zopheridae, followed by Ptinidae, Hybosoridae, Chrysomelidae, Nitidulidae, and Elateridae (Figure 5A). As shown in Figure 5B, the Interval Plot of the pooled standard deviations revealed the three highest waves of beetles’ succession distinctively to the snake-envenomized corpses represented by Histeridae, followed by Dermestidae and Scarabaeidae, followed by another five waves to the control ones by Dermestidae, followed by Histeridae, Scarabaeidae, Curculionidae, and Tenebrionidae. Only three moderate waves, represented by Histeridae, followed by Dermestidae and Scarabaeidae, were attracted to the scorpion-envenomized corpses. These data clearly show that Dermestidae, Scarabaeidae, and Histeridae were the three prevalent families attracted to all corpses, while Curculionidae and Tenebrionidae were distinctively attracted to control corpses only. The beetles were observed to be less attracted to envenomized corpses, with scorpion-envenomized corpses attracting less families compared to the snake-envenomized ones (Figure 4).

The overall Kruskal–Wallis test revealed significant differences in the mean number of attracted beetles within treatments (α < 0.05, H = 182.74, DF = 41, *p* < 0.05). As shown in Figure 5A, the Mann–Whitney *U*-test revealed that snake-envenomized corpses attracted significantly higher numbers of beetles from Anthicidae and Staphylinidae but a significantly lower number of Cleridae and Curculionidae compared to the control ones (*p* < 0.05, *n* = 5). However, scorpion-envenomized corpses attracted significantly higher numbers of beetles of Histeridae and Staphylinidae but a lower number of Curculionidae and Tenebrionidae compared to the control ones. When comparing the envenomized corpses, the snake-envenomized ones attracted significantly higher numbers of Anthicidae, Curculionidae, and Tenebrionidae but a significantly lower number of Cleridae compared to scorpion-envenomized ones (*p* < 0.05, *n* = 5). The control corpses distinctively attracted Chrysomelidae, Elateridae, and Hybosoridae. A variation in the abundance of beetle families was observed between venom treatments, with the snake-envenomized corpses attracting higher numbers than the scorpion-envenomized ones.

### 3.6. Differential Succession of Beetles

A total of 27 species of beetles belonging to 14 families were reported and identified in this study (Table 3). Of them, Anthicidae was represented by 38 specimens of *Omonadus formicarius* (Goeze, 1777); Chrysomelidae was represented by 4 *Caryedon acaciae* (Gyllenhal, 1833); Cleridae was represented by 23 *Necrobia rufipes* (Fabricius, 1781) and 8 *Necrobia* sp.; Curculionidae was represented by 2 *Coccotrypes rhizophorae* (Hopkins 1915) and 57 *Dinoderus* sp.; Dermestidae was represented by 9 *Attagenus posticalis* Fairmaire, 1879, 127 *Dermestes maculatus* De Geer, 1774, and 114 *Dermestes frischi* Kugelman, 1792; Elateridae was represented by 3 *Aeoloides grisescens* (Germar, 1844); Histeridae was represented by 294 *Saprinus chalcites* (Illiger, 1807) and 4 *Saprinus caerulescens* (Hoffmann, 1803); Hybosoridae is represented by 3 *Hybosorus illigeri* Reiche, 1853; Nitidulidae was represented by 8 *Carpophilus hemipterus* (Linnaeus, 1792) and a singleton (1 individual beetle) of *Urophorus humeralis* (Fabricius, 1798); Ptinidae was represented by 3 *Stegobium paniceum* (Linnaeus, 1758); Scarabaeidae was represented by 181 *Aphodius adustus* Klug, 1855, 27 *Malader insanabilis* (Brenske, 1894), and 10 *Rhyssemus saoudi* Pittino, 1984; Staphylinidae was represented by 4 *Leptacinus* sp. and 62 *Philonthus* sp.; Tenebrionidae was represented by a singleton of *Adesmia cancellate* Klug,1830, 11 *Alphitobius diapernius* (Pancer, 1797), 22 *Mesostena pincticollis* Solier, 1835, 45 *Opatroides punctulatus* Brulle, 1832, and 31 *Thriptera crinite* (Klug, 1830); and Zopheridae was represented by a doubleton (2 individual beetles) of *Synchita* sp. (Table 3).

Figure 6 represents the differential succession of the 14 coleopteran families during the four decomposition stages. Out of them, eight families were attracted to all corpses in various succession manners (Figure 6A,C–E,G,K–M). The Kruskal–Wallis test revealed an overall significant difference in the mean number of the attracted beetle families within the treatments (α < 0.05, H = 182.74, df = 41, *p* < 0.05). The non-parametric Mann–Whitney *U*-test revealed that the control corpses attracted significantly more beetles from Curculionidae, Histeridae, Nitidulidae, and Staphylinidae during the decay and/or dry stages compared to the envenomized ones (*p* < 0.05, *n* = 5) (Figure 6C,D,K,M). For comparing the envenomized corpses, the snake-envenomized ones attracted significantly more beetles from Anthicidae, Dermestidae, and Histeridae during the decay stages compared to the scorpion envenomized ones (*p* < 0.05, *n* = 5) (Figure 6A,E,G). While scorpion-envenomized corpses attracted significantly more beetles from Cleridae and Histeridae during the dry stages compared to the snake-envenomized ones (*p* < 0.05, *n* = 5) (Figure 6C,G).

The other six families were attracted to corpses in a selective manner. This was evidenced by Chrysomelidae, Elateridae, and Hybosoridae, which were distinctively attracted to the control corpses only during most of the decomposition stages (Figure 6B,F,H). Nitidulidae, Ptinidae, and Zopheridae were distinctively attracted to the control and snake-envenomized corpses only during the decay and dry stages (Figure 6I,J,N). None was distinctively attracted to the scorpion-envenomized corpses. These beetle families exhibited selective attraction patterns, differing by venom treatment and specific stages of decomposition.

At the species level, the data in Table 3 represents venom-, stage-, and temporal-based differential succession of corpse-attracted beetle species during the decomposition process. These data reveal a distinctive association to the control corpses by *C. acaciae* (Chrysomelidae), followed by *H. illigeri* (Hybosoridae), *A. grisescens* (Elateridae), and *C. rhizophorae* (Curculionidae), starting from the fresh stage (within 31 h postmortem). *O. formicarius* (Anthicidae), *S. chalcites* (Histeridae), *A. adustus* (Scarabaeidae), and *O. punctulatus* (Tenebrionidae) were predominantly associated with all the experimental corpses, starting from the fresh stage (within 31 h postmortem). There was a singleton of each of *A. cancellate* (Tenebrionidae) and *U. humeralis* (Nitidulidae) reported distinctively associated with the snake-envenomized corpses during the fresh stage (within 21 h) and decay stage (within 31 h) postmortem, respectively. A singleton of *Synchita* sp. (Zopheridae) was distinctively associated with the control and snake-envenomized corpses during the decay stages within 45 and 31 h postmortem, respectively. Finally, a double ton of *C. rhizophorae* (Curculionidae) was distinctively associated with the control corpses during the fresh stage (within 31 h postmortem). 

## 4. Discussion

The present study proposed a forensic scenario for the envenomation-related death and its impact on the postmortem decomposition process and the succession pattern of corpse-attracted beetles. The forensic relevance of envenomation-related deaths is highlighted by the substantial global and regional incidence of venomous animal bites and stings. According to the WHO, snakebite envenoming causes tens of thousands of deaths annually worldwide and leads to significant morbidity [34]. Scorpionism also contributes notably to mortality and morbidity, particularly in arid regions like Saudi Arabia [30,42,43]. Regional epidemiological studies report significant numbers of snakebite and scorpion sting cases in Saudi Arabia, with thousands of annual incidents [41,42]. This epidemiological context justifies the critical need for detailed forensic investigations into how envenomation influences decomposition and insect succession, which can aid in accurate postmortem interval estimation and cause-of-death differentiation. It is important to clarify four points: (a) neurotoxic venoms from two different local venomous animals, the *W. aegyptia* snake and the *A. crassicauda* scorpion, have been targeted to achieve the goal of this study; (b) the venoms were freshly extracted from the animals, and their lethality was determined prior to carrying out this study; (c) although domestic pigs are preferably used as a human analog in forensic experiments [86], using pigs for experimental purposes is prohibited in Saudi Arabia, and, thus, we alternatively used rabbits as experimental models in the present study; (d) the experimental rabbits were envenomized by intravenous injection with the lethal doses of venoms aliquots rather than being directly bitten or stung by the snake or scorpion..

Moreover, it is important to clarify that the botanical garden was targeted as the experimental site for conducting this study, as it shares several ecological features with rural agricultural zones, which are globally recognized as high-risk settings for envenomation incidents. According to the WHO’s estimations [34], over 80% of snakebite cases worldwide occurred in rural agricultural areas. Similarly, Chippaux (2017) [87] emphasized that, in Africa and the Middle East, most snakebites occur during agricultural activities such as harvesting and irrigation. Additionally, Forrester et al. (2018) [28] reported that the majority of animal-related fatalities in the United States (2008–2015) occurred in rural and agricultural settings, often during fieldwork. Therefore, we believe that the semi-natural, plant-rich environment used in this study may reflect the ecological realities encountered in field-based forensic scenarios. This further supports the use of this experimental site as a forensically relevant model, particularly for envenomation-related deaths in rural settings.

It is well-known that the process of corpses’ decomposition constitutes a crucial step in the recirculation of organic materials throughout the food chain in the nutrition cycle [78,88]. From the forensic point of view, this process provides vital evidence in the legal investigations, as it has unique spatial- and temporal-dependent successive stages that aid the investigators in determining the PMI [89,90]. This process usually takes place in 4–5 stages [8], depending on many interacting environmental and non-environmental factors. The environmental factors include the atmospheric temperature and humidity [91,92], the habitat [80,93,94], and the type of soil in scene [95]. The non-environmental factors include the corpse’s chemical composition [7], microbial composition and activity [96,97], the activity of the colonizing insects [98], the barriers such as clothes and coverages [99,100], and the cause of death [49,72,101,102].

Based on the aforementioned parameters, and in order to avoid any possible discrepancies, the current study was carried out in a botanical garden with a homogeneous clayey sandy soil, and all experimental groups of corpses were placed in identical metal cages in similar shady places away from direct sunlight. Furthermore, this study was conducted in June, which is a hot and dry month characterized by stable climatic conditions, with normal atmospheric averages of temperature, humidity, and wind speed of this time of the year in Riyadh city [45]. Thus, through the 11-day experimental duration, the corpses reached the dry stage within 4 to 5 days in these atmospheric conditions. Therefore, we expect that there was no external climatic impact on the decomposition process and beetle succession in this study. We hypothesize that the reported variation in the abundance and succession of beetles between the treated corpses may be attributed to the antemortem envenomization. Consequently, the current study reported four main successive decomposition stages: fresh, bloating, decaying, and dried, as per reported in our previously published works [46,47,49].

Envenomation by venomous animals is a considerable global cause of mortality [28,29,33,103,104]. In the Middle East and Africa, thousands of snakebites, as well as scorpion and spider stings, were documented [16,30,33,105,106]. Yet, few studies have investigated the decomposition process and succession of forensic insects on corpses upon death by envenomation with scorpion venoms [102,107] and snakes [49,52,108]. To the best of our knowledge, no such studies have been conducted in Saudi Arabia, and hence, the present study was undertaken to address this gap. Our data showed clearly that envenomization with snake and scorpion venoms both reduced the duration of the fresh stage and elongated that of the decay stage. The reported significant variation in lethality between the *W. aegyptia* snake and the *A. crassicauda* scorpion has been reflected as a variation in the rate of decomposition. This may indicate that envenomization has shortened the duration of the fresh stage, as the bloating stage started earlier compared to the control. The prolonged decomposition rate was more pronounced in the scorpion-envenomized corpses in terms of longer duration in the bloating and decay stages. Consequently, the overall duration of the decomposition process was elongated in terms of a delay (up to one day) at the start of the dry stage compared to that of the snake-envenomized ones. In contrary, our previously published work [49] reported an acceleration in the decomposition rate of rabbit corpses upon envenomization with venoms from the Egyption *Naja haje* and *Cerastes cerastes* snakes. These findings may indicate that the postmortem decomposition rate is venom-dependent and, hence, could lead to bias in the estimation of PMI. There is evidence for this conclusion provided by other studies that investigated how intoxication with different types of toxic substances impacted postmortem carcass decomposition and insect succession differently [54,72,101,109]. Comprehending these venom-dependent phenomena is essential for forensic entomologists, as it facilitates the enhancement of PMI approximations by considering the particular venom variant implicated.

The levels of lethality of the *W. aegyptia* snake (from 0.170 to 0.180 mg/kg) [63,110] and the *A. crassicauda* scorpion (from 1.1 to 1.7 mg/kg) have been estimated differently in different studies for [64,70]. These variations could be attributed to the variation in the laboratory and experimental conditions between the investigators and to many other factors [111,112]. Our data revealed similar lethality (1.4 mg/kg) to that previously reported. While it was around 3.3 times greater than that determined by Al-Sadoon’s group for the snake venom, this difference could be attributed to the variation in the injection routes and in both the type and body weight of the experimental animals and other factors [64,113]. In this regard, Al-Sadoon and his colleagues used the intraperitoneal route in rats (weighing 200–250 g each), while we used the subcutaneous route in mice (weighing 18–20 g each) [111]. Our data also determined the snake venom lethality of 26.7 times that of the scorpion, which could be attributed to its chemical components, as it contains many different kinds of enzymes [114,115] and non-enzymatic proteins [116]. Consequently, it has multiple modes of action, mainly cardio and neurotoxicity, interruption of many vital systemic functions [63,110,117,118,119], and, finally, systemic dysfunction leading to death [120]. The *A. crassicauda* scorpion venom contains smaller neurotoxin polypeptides of low molecular weight, simple proteins with lethal neurotoxic, and paralytic effects [70], resulting in cardio-respiratory failure and eventually death [120,121]. This variation in the chemical composition and mode of action may explain the concomitant variation in the decomposition process, as the scorpion-envenomized corpses decomposed slower than the control and snake-envenomed ones. In contrary, other causes of death, like antemortem heroin-injection, was found to accelerate the decomposition of rabbit corpses [72]. These findings may suggest that the antemortem cause of death could potentially impact the postmortem decomposition process and that envenomation impacts differently the decomposition process based on the type of venom.

It is well known that dipteran flies are the first to attract to corpses during the very early decomposition stages, while others, like coleopteran beetles, are usually attracted to corpses during the later stages [8]. We have investigated the successional pattern, species abundance, corpse colonization, and development of dipteran flies’ immature stages (comparing field-collected versus lab-reared ones) upon rabbits’ envenomization. On the other hand, beetles have not received as much attention in research—especially when it comes to how venom affect certain outcomes. Thus, the current study investigated how snake and scorpion venoms affect their successional pattern as corpse decomposition progresses. Our data showed a maximum abundance of corpse-attracted beetles between days 4 and 5 (during the decay stage). This unique pattern of insect succession could be due to the effect of the unique cadaveric volatile organic components that give cadavers their unique smell and attract a wide range of cadaver-seeking insects. In this context, up to 104 cadaveric volatile chemical compounds were identified during the decomposition process [122]. Therefore, the stronger and more unpleasant smell coming from scorpion-envenomized corpses, compared to those of the other groups, may be attributed to the effect of certain chemicals in the scorpion venom [64,66]. Evidence for this is provided by [122], who reported cadaveric volatile components differently in different biotopes. This may explain the attraction of the low number of beetles to the scorpion-envenomized corpses compared to those of the control and snake-envenomized ones. Moreover, the delay in the abundance of the attracted beetles on the scorpion-envenomized corpses (up to day 5), compared to the snake-envenomed ones, may be attributed to the difference in the chemical components and the mode of action of venoms. A recent study conducted by [52] reported fewer attracted beetles to envenomed carcasses compared to controls and fewer attracted beetles to *N. haje* snake envenomed carcasses compared to *C. cerastes* snake-envenomed ones. However, other causes of death, like heroin injection, showed no impact on the postmortem beetles’ succession patterns [72].

The data of differential abundance revealed a variation in the succession of the coleopteran families. The most predominant corpse-attracted family was Histeridae (298 beetles) during all decomposition stages except the fresh stage. The second was Dermestidae (250 beetle), followed by Scarabaeidae (218 beetles), both during all stages, except for the fresh and bloating ones. The third was Tenebrionidae (110 beetles) during the selective decomposition stages. The least was Zopheridae (2 beetles) during the decay stage. In agreement with our results, Dermestidae, Histeridae, and Scarabaeidae were also reported as predominant families associated with rabbit corpses [80,123] and with pig corpses [124]. These data suggest two differential succession manners. The first is a differential predominant manner shown by four families (Curculionidae, Histeridae, Nitidulidae, and Staphylinidae), three families (Anthicidae, Dermestidae, and Histeridae), and two families (Cleridae and Histeridae), which were particularly predominant on the control, snake-envenomized, and scorpion-envenomized corpses, respectively, during the decay and/or dry stages. Histeridae was the only predominant family on all corpses, regardless of the type of treatment. The second manner is a distinctive predominant manner shown by three families (Chrysomelidae, Elateridae, and Hybosoridae), which were distinctively attracted to control corpses only, and three families (Nitidulidae, Ptinidae, and Zopheridae), which were attracted to both control and snake-envenomized corpses only. No definitive families were distinctively attracted to scorpion-envenomized corpses. These differentially attracted beetles could serve as potential indicators for differentiating between the types of venoms and, consequently, suggest a venom-dependent succession manner. On the other hand, the unexpected presence of Chrysomelidae, Curculionidae, Elateridae, and Zopheridae on corpses provides insight into the intricate ecological interactions surrounding the corpses. These particular families are known primarily for their plant-feeding or wood-boring behaviors rather than being necrophagous [125]. This may be attributed to either the surrounding vegetation of the experimental site or their opportunistic feeding on fungi growing on corpses. This, in fact, may suggest a broader ecological context for the decomposition process, a potential impact on the succession pattern of more conventional necrophagous insects and ecological indication of the crime scene, which could be of forensic relevance. In forensic investigations, the accurate identification of particular species of beetles may assist in ascertaining whether a fatality resulted from envenomation by a snake or a scorpion, thereby offering critical evidence in circumstances where the etiological factors of death remain ambiguous.

At the species level, *C. acaciae* (Chrysomelidae) was distinctively associated with control corpses only during all decomposition stages except the bloating stage. *O. formicarius* (Anthicidae), *S. chalcites* (Histeridae), *A. adustus* (Scarabaeidae), and *O. punctulatus* (Tenebrionidae) were predominantly associated with all corpses during most decomposition stages. These data may indicate an association of a certain beetle species with the treatment but not with the decomposition stages. Furthermore, the reported singleton and doubleton were associated with the fresh and decay stages, which may suggest an association of the singleton or doubleton with particular decomposition stages of a particular treatment. This is in contrary to the findings of [80], who reported no association between the attraction of a single species and a particular stage of decomposition. This may be due to the difference in the experimental sites, as they were comparing different habitats (agricultural, desert, and urban). No definitive species was distinctively reported as associated with the scorpion envenomized corpses. Reporting certain corpse-associated beetles during the early stages of decomposition, like the fresh and bloating stages, might be due to their seasonal appearance rather than linking to the decomposition stage [126].

Finally, it is important to acknowledge that the use of rabbit carcasses, although practical and ethically manageable, presents limitations when extrapolating findings to human forensic cases. This is because the substantial difference in body mass between rabbits and humans may influence both decomposition rate and insect succession dynamics. Previous studies have demonstrated that body size significantly affects postmortem changes, with larger carcasses exhibiting prolonged decomposition stages and altering insect colonization patterns [51,127,128,129]. While rabbits provide a consistent and accessible model for preliminary entomological research, future studies incorporating larger animal models such as pigs, or actual human remains when ethically and legally permissible, are essential to validate and expand upon our findings.

## 5. Conclusions

This study revealed that antemortem envenomization of rabbits with snake and scorpion venoms has significantly affected the corpses’ decomposition rate, with the impact being more pronounced in scorpion-envenomized ones. This impact was evidenced by the elongation of the bloating and decay stages, as well as a delay of up to one day in reaching the dry stage compared to the control corpses. The differential abundance and succession pattern of corpse-associated beetles varied significantly between the envenomized and control corpses, suggesting a venom-dependent succession. Notably, families such as Histeridae, Dermestidae, Scarabaeidae, and Tenebrionidae were predominant in all corpses. Others showed distinctive associations with specific treatments or decomposition stages. At the species level, there were unique associations between certain beetle species and specific treatments or decomposition stages. For instance, the singleton of *U. humeralis* (Nitidulidae) and *A. cancellate* (Tenebrionidae) were distinctively associated with snake-envenomized corpses. The doubleton of *Synchita* sp. (Zopheridae) was distinctively associated with the control and snake-envenomized corpses. *C. acaciae* (Chrysomelidae) and *A. grisescens* (Elateridae), followed by a doubleton of *C. rhizophorae* (Curculionidae) and *H. illigeri* (Hybosoridae), were distinctively associated with control corpses only. These findings may indicate that the succession of the reported beetles is venom-dependent, which could be helpful as envenomation markers. This may aid in understanding the influence of various venom types of various lethality from various venomous animals on both corpse decomposition and insect succession, as well as identifying which venom type has the most significant postmortem impact. Finally, the herbivorous (non-necrophagous) families, Chrysomelidae, Curculionidae, Elateridae, and Zopheridae, were unexpectedly reported to be associated with corpses, which could be potential ecological markers of the crime scene. This, in fact, may provide additional valuable forensic evidence in forensic investigation and updating the database of the envenomation-related corpse decomposition process and the associated beetle taxa in Saudi Arabia. These findings can enhance PMI estimation and aid in differentiating between causes of death involving venomous animals, ultimately contributing to more accurate crime scene reconstructions and legal investigations.

## Figures and Tables

**Figure 2 insects-16-00625-f002:**
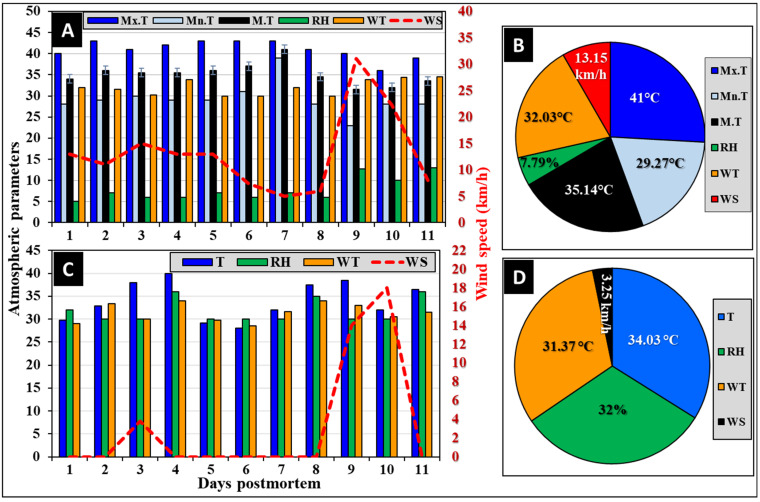
The meteorological parameters over the experimental period show the atmospheric parameters (**A**) and their overall averages (**B**) in Riyadh city and the on-site manually recorded daily weather parameters (**C**) and their overall averages (**D**) at the experimental sites. Mx.T: maximum temperature; Mn.T: minimum temperature; M.T: the mean of maximum and minimum temperatures (*n* = 11); RH: relative humidity; WT: wind temperature; and WS: wind speed.

**Figure 3 insects-16-00625-f003:**
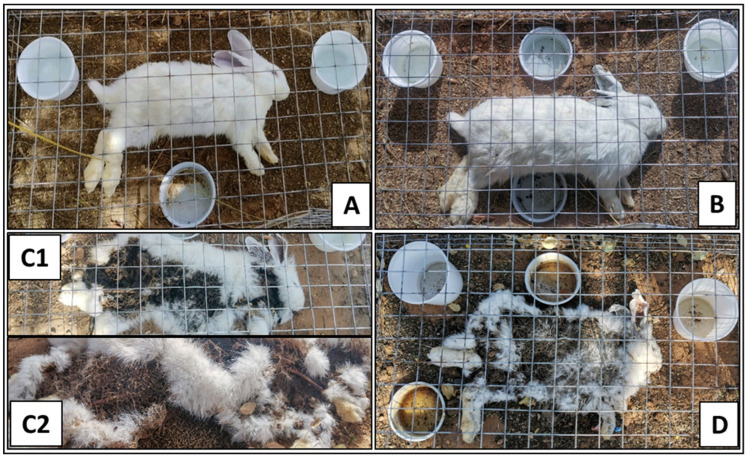
The postmortem characteristic features of the reported four decomposition stages of the rabbit corpses inside their metal cages. (**A**): Fresh stage, (**B**): bloating stage, (**C1**,**C2**): decayed stage and the associated colonizing larvae, respectively, and (**D**): dry stage.

**Figure 4 insects-16-00625-f004:**
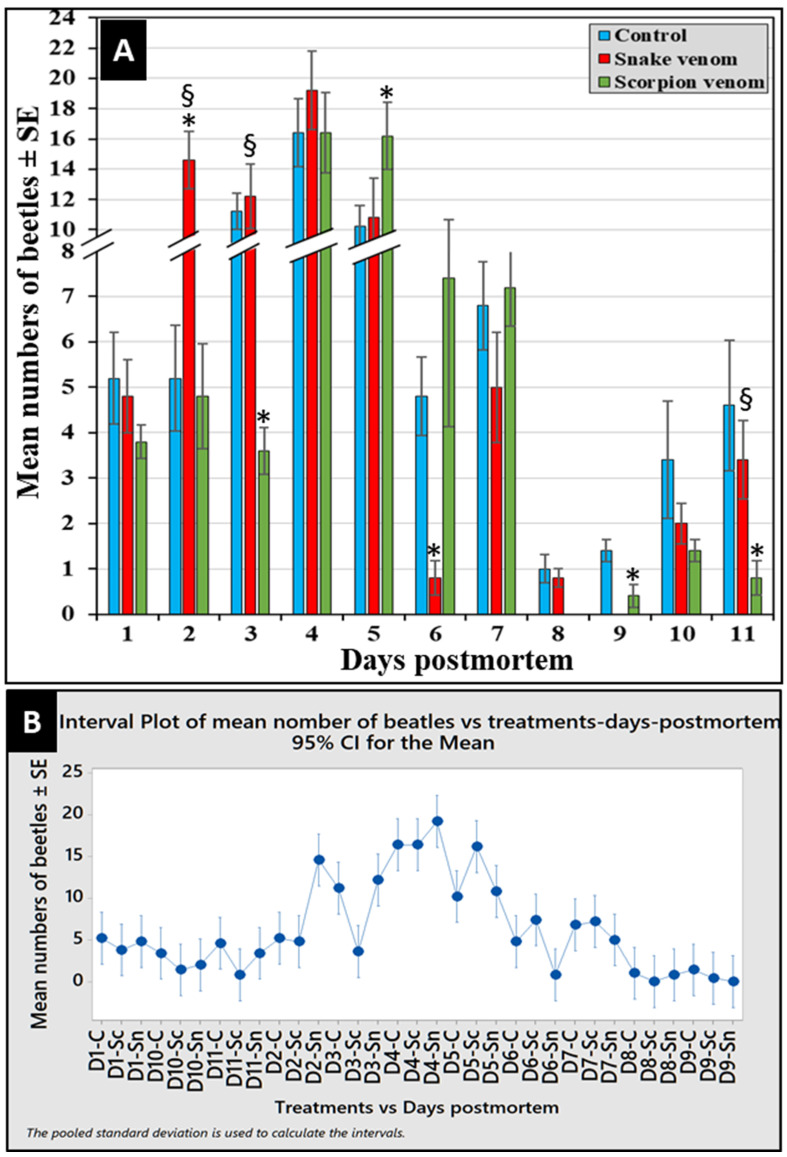
The abundance of corpse-attracted beetles post-envenomization (**A**) and the Interval Plot of mean numbers of beetles (**B**) in each treatment versus days over the experimental period (from D1 to D11). C: control; Sn: snake-envenomized; Sc: scorpion-envenomized corpses. The error bars represent standard errors of means of five replicates (*n* = 5). The (*) symbol indicates significant differences compared to the control, while the (§) symbol indicates significant differences between envenomized corpses (*p* < 0.05, *n* = 5).

**Figure 5 insects-16-00625-f005:**
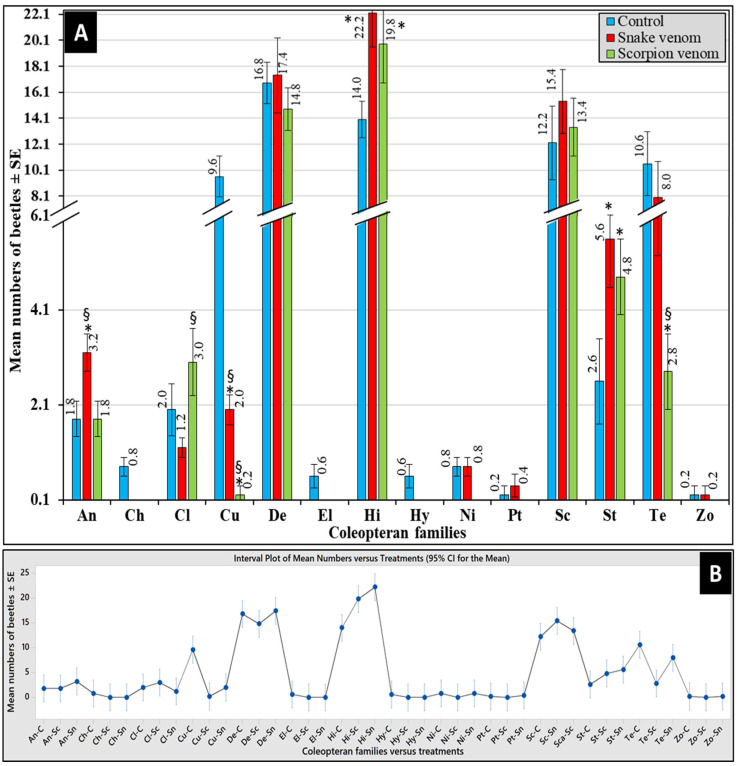
The abundance of the reported corpse-attracted coleopteran families post-envenomization over the experimental period (**A**). The error bars represent the standard errors (SE) of means of five replicates (*n* = 5). The Interval Plot of the pooled standard deviations (**B**) shows various succession waves of coleopteran families versus envenomization with snake-envenomized (Sn), scorpion-envenomized (Sc), and control (C). An: Anthicidae; Ch: Chrysomelidae; Cl: Cleridae; Cu: Curculionidae; De: Dermestidae; El: Elateridae; Hi: Histeridae; Hy: Hybosoridae; Ni: Nitidulidae; Pt: Ptinidae; Sc: Scarabaeidae; St: Staphylinidae; Te: Tenebrionidae; Zo: Zopheridae. The (*) symbol indicates significant differences compared to the control, while the (§) symbol indicates significant differences between envenomized corpses (*p* < 0.05).

**Figure 6 insects-16-00625-f006:**
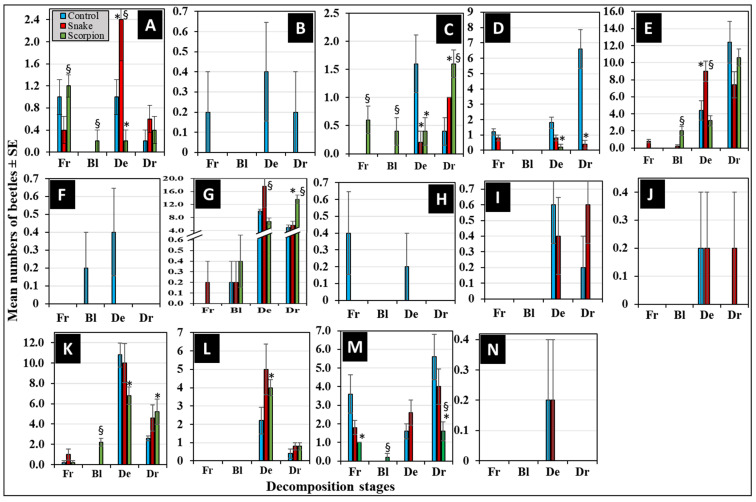
Differential abundance of corpse-attracted coleopteran families during the decomposition stages: fresh (Fr), bloating (Bl), decayed (De), and dry (Dr). The individual figures represent the Anthicidae (**A**), Chrysomelidae (**B**), Cleridae (**C**), Curculionidae (**D**), Dermestidae (**E**), Elateridae (**F**), Histeridae (**G**), Hybosoridae (**H**), Nitidulidae (**I**), Ptinidae (**J**), Scarabaeidae (**K**), taphylinidae (**L**), Tenebrionidae (**M**), and Zopheridae (**N**) families. The (*) symbol indicates significant difference compared to the control, while the (§) symbol indicates significant difference between the envenomized corpses (*p* < 0.05, *n* = 5).

**Table 1 insects-16-00625-t001:** Results of Probit analysis showing LD_50_ and LD_95_ of snake and scorpion venoms in mice at 24 h post-envenomization.

Venom Types	LD_50_ (mg/kg) (Lower–Upper)	LD_95_ (mg/kg) (Lower–Upper)	Slope ± SE
*W. aegyptia*	0.053 * (0.052–0.054)	0.066 ** (0.063–0.069)	18.29 ± 3.22
*A. crassicauda*	1.416 * (1.288–1.558)	2.516 ** (2.251–2.813)	6.59 ± 0.91

Values marked by * and by ** are significantly different (*p* < 0.5) as their 95% confidence limits (lower to upper) are not overlapped, according to [81,85].

**Table 2 insects-16-00625-t002:** The duration of the decomposition stages of the envenomized corpses post-envenomization with snake and scorpion venoms.

Types of Venoms	Days Postmortem																	
	1		2				3	4			5		6	7	8	9	10	11
Control																		
*W. eagiptia*																		
*A. crassicauda*																		
	Keys		Fresh stage							Bloated stage				Decay stage			Dry stage	


**Table 3 insects-16-00625-t003:** Differential attractions of coleopteran families and species during the decomposition process post-envenomization over the experimental period.

Coleopteran Families (Total Number)	Beetle Species (Total Number)	Treatments	Control				*W. aegyptia*				*A. crassicauda*			
		DS	Fr	Bl	De	Dr	Fr	Bl	De	Dr	Fr	Bl	De	Dr
		HPM *	0–31	31–45	45–93	93→	0–21	21–31	31–93	93→	0–21	21–45	45–117	117→
Anthicidae (38)	*O. formicarius* (38)		+	−	+	+	+	−	+	+	+	+	+	+
Chrysomelidae (4)	*C. acaciae* (4)		+	−	+	+	−	−	−	−	−	−	−	−
Cleridae (31)	*N. rufipes* (23)		−	−	+	+	−	−	+	+	−	−	+	+
	*Necrobia* sp. (8)		−	−	+	−	−	−	−	+	+	+	−	−
Curculionidae (59)	*Dinoderus* sp. (57)		+	−	+	+	+	−	+	+	−	−	+	−
	*C. rhizophorae* (2)		+	−	−	−	−	−	−	−	−	−	−	−
Dermestidae (250)	*D. maculatus* (127)		−	−	+	+	−	+	+	+	−	+	+	+
	*D. frischi* (114)		−	−	+	+	−	−	+	+	−	+	+	+
	*A. posticalis* (9)		−	−	+	+	+	−	+	−	−	−	+	−
Elateridae (3)	*A. grisescens* (3)		−	+	+	−	−	−	−	−	−	−	−	−
Histeridae (298)	*S. chalcites* (294)		−	+	+	+	+	+	+	+	−	+	+	+
	*S. caerulescens* (4)		−	−	−	−	−	−	+	−	−	−	+	+
Hybosoridae (3)	*H. illigeri* (3)		+	−	+	−	−	−	−	−	−	−	−	−
Nitidulidae (9)	*C. hemipterus* (8)		−	−	+	+	−	−	+	+	−	−	−	−
	*U. humeralis* (1)		−	−	−	−	−	−	+	−	−	−	−	−
Ptinidae (3)	*S. paniceum* (3)		−	−	+	−	−	−	−	+	+	−	−	−
Scarabaeidae (218)	*A. adustus* (181)		+	−	+	+	+	−	+	+	−	+	+	+
	*R. saoudi* (10)		−	−	+	+	+	−	+	+	−	+	+	+
	*M. insanabilis* (27)		−	−	+	+	+	−	+	+	+	+	+	+
Staphylinidae (66)	*Philonthus* sp. (62)		−	−	+	+	−	−	+	+	−	−	+	+
	*Leptacinus* sp. (4)		−	−	−	−	−	−	+	−	−	−	−	−
Tenebrionidae (110)	*M. pincticollis* (22)		+	−	+	+	+	−	−	+	+	−	−	−
	*T. crinite* (31)		+	−	+	+	+	−	−	+	+	−	−	+
	*A. diapernius* (11)		−	−	−	+	−	−	−	−	−	−	−	−
	*O. punctulatus* (45)		+	−	+	+	+	−	+	+	+	+	−	+
	*A. cancellate* (1)		−	−	−	−	+	−	−	−	−	−	−	−
Zopheridae (2)	*Synchita* sp. (2)		−	−	+	−	−	−	+	−	−	−	−	−

Ds: decomposition stages; Fr: fresh; (Bl): bloating; (De): decayed; (Dr): dry; HPM: hours postmortem; *: start-to-end hours (duration) postmortem of each decomposition stage. The arrow (→) indicates extended duration. The positive sign (+) indicates the recorded beetle, while the negative sign (−) indicates no record.

## Data Availability

The original contributions presented in this study are included in the article. Further inquiries can be directed to the corresponding author.

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
