# Peer review of "Death-Leading Envenomization of Rabbits with Snake Versus Scorpion Venoms: A Comparative Forensic Investigation of Postmortem Decomposition and Beetle Succession"

_insects, 2025, doi:10.3390/insects16060625_

Round 1

Reviewer 1 Report

Comments and Suggestions for Authors

This manuscript focuses on the study of the effects of snake and scorpion venoms on body decomposition and insect succession. I have reviewed a previous version of this manuscript, and the new version has shown significant improvements in quality. The following are the suggestions for revising the manuscript:

Lines 32 - 33: The original text states, “A total of 1094 beetles belonging to 27 species of 14 families were reported.” Compared with the six species found on the carcasses in the previous version, the current discovery of 27 species indicates an enhanced systematic nature of the study. However, the authors need to carefully distinguish the relevance of these species to the carcass ecosystem, clarifying which species are attracted to the carcass for feeding and which merely appear on the carcass accidentally. Take the species of Chrysomelidae as an example; most of them are herbivorous insects. Their presence on the carcass is likely due to the availability of suitable plants for feeding near the carcass rather than being attracted by the carcass itself. Therefore, the authors should carefully screen and exclude the species that do not belong to the carcass ecosystem from both the abstract and the research results.

Lines 56 - 63: The manuscript mentions “The corpse - seeking insects, which are categorized into four groups.” However, the theoretical basis for this classification method is not clearly explained. After reading the book by Byrd and Tomberlin cited by the authors, no relevant classification content was found. The authors are requested to supplement reasonable literature citations to support the scientificity and rationality of this classification.

Line 64: It is recommended to change “bacteria” to “microorganisms.” Fungi also play a crucial role in the process of corpse decomposition.

Line 238: There is no full - stop at the end of this sentence. It is recommended to add the punctuation mark to ensure the standardization of the text format.

Line 480: Table 3 presents the succession patterns of different insects on the corpse, which is an important indicator for estimating the post-mortem interval in forensic practice. To enhance the practicality of the data, it is suggested that the authors present the daily occurrence of each species in detail. For example, the contents such as Fr, Bl, De, and Dr should be changed to 1, 2, 3…11d. Considering that the revised table may be too large, the tables for different treatments can be divided into three parts. In addition, since the authors used five carcasses as replicate samples, they need to provide a detailed description of how these data were processed, including the differences in beetles among the five carcasses and the analytical and processing methods adopted to address these differences.

Discussion Section: The authors selected rabbits, which have a significant size difference from human corpses, as the animal model. Previous studies have demonstrated that the size of the remains affects body decomposition and insect succession and have compared the differences in decomposition and insect succession between different animal models and human corpses. The authors are advised to refer to the following literature to thoroughly explore the potential limitations of using rabbits as a model animal in forensic practice:

Matuszewski, K. Fratczak, S. Konwerski, D. Bajerlein, K. Szpila, M. Jarmusz, M. Szafalowicz, A. Grzywacz, A. Madra, Effect of body mass and clothing on carrion entomofauna, Int. J. Legal Med. 130 (2016) 221–232.

Matuszewski, S. Konwerski, K. FraË›tczak, M. SzafaÅ‚owicz, Effect of body mass and clothing on decomposition of pig carcasses, Int. J. Legal Med. 128 (2014) 1039–1048.

Wang, M. Ma, X. Jiang, J. Wang, L. Li, X. Yin, M. Wang, Y. Lai, L. Tao, Insect succession on remains of human and animals in Shenzhen, China, Forensic Sci. Int. 271 (2017) 75–86.

Reviewer 2 Report

Comments and Suggestions for Authors

The subject area addressed by the present research is an interesting, albeit narrow, topic of significance to forensic entomology. The work is presented under the premise that envenomation-related deaths are significant globally, yet the authors do not provide any data to support the claim. It would strengthen the Introduction and Discussion to provide global or regional numbers to support statements made in the manuscript. Overall, the methodology is appropriate. However, the authors should explain why insect colonization as a whole, as opposed to beetles, was not examined in this study. As the authors acknowledged, flies are the most significant colonizers of animal remains. Their pattern of succession does influence subsequent beetle attraction to human and animal remains, yet the flies were not addressed as a potential factor influencing the data of this study. Without considering fly colonizers on the envenomated rabbits, the beetle succession cannot be fully evaluated. To treat beetle colonization as a separate, unrelated process is a major weakness of the paper. 

Specific comments follow:

The manuscript could use some editing in terms of sentence structure and word choices.

  1. Line 54. Secrets is not a good word choice
  2. Lines 56-58. There are other coleopteran families that are necrophagous besides Dermestidae.
  3. Lines 58-60. The two families listed are not the only ones that are attracted to gut contents.
  4. Lines 73-74. Silphidae?
  5. Line 83. Spiders are arthropods.
  6. Lines 97-104. Temperature was not continuously monitored?
  7. Lines 108-113. The study site does not seem representative of a typical location in Saudi Arabia. Do the authors feel that insect fauna is representative of what would occur at a crime scene?
  8. Lines 170-181. Technically, a fourth group should have been used as a negative control in which no treatment (i.e., no saline injection) was used on the rabbits.
  9. Section 2.8. The authors indicate that four stages of decomposition were examined, yet mention 5 in the Introduction and referenced them by different names. Why 4 stages versus 5?
  10. Section 2.9. Provide details of beetle sampling. For example, were all beetles observed on the rabbits collected or only some? If the beetles were removed during observation periods, how might that have impacted the rate of decomposition and subsequent visitation/colonization by other beetles?
  11. Provide references for the identification keys used to beetle identifications. Were voucher specimens made from this study?
  12. Results section. In several places, the authors have provided interpretations of data. Be sure to avoid doing so in the results; save the interpretations for the Discussion.
  13. Lines 305-326. Were the durations of each stage of decomposition under different treatments statistically compared? The statistical measures are not presented in this section.
  14. Line 310. By larvae, do the authors mean beetle or fly larvae?
  15. Section 3.6. Be sure to write in the past tense when reporting data.
  16. Lines 451 and after. When reporting statistical outcomes, provide the statistical measures and not just the p value.
  17. Line 527. global death-leading cause is an awkward statement that should be reworked.
Comments on the Quality of English Language

The manuscript could use some editing in terms of sentence structure and word choices.

Round 2

Reviewer 1 Report

Comments and Suggestions for Authors

Thank the authors for their careful replies and revisions!

Regarding Table 3, the authors should have a clear understanding of the theoretical basis of forensic entomology for estimating the postmortem interval (PMI) based on the succession pattern. Generally, forensic entomologists estimate the PMI by considering the coexistence time of different insects on the corpse. For instance, if species A is present on the corpse within a time range of 1 to 3 days, and species B is present within a time range of 2 to 4 days, then the corresponding time interval when both species are observed coexisting on a corpse is 2 to 3 days. If the authors categorize it according to different stages of corpse decomposition, this approach fails to reflect the significance of the succession pattern in estimating the PMI, because the accuracy is extremely poor in this way. Figures 4 and 5 lack detailed information about each individual species, which significantly undermines the resolution of the data presented. Therefore, from the perspective of practical applications in forensic entomology, I believe it would be more advisable to use data with a daily unit. Alternatively, whether this part needs to be modified can also be left to the academic editor for review and decision.

Author Response

Please, find the attached file 
